# Anemia and its associated factors among women of reproductive age in eastern Africa: A multilevel mixed-effects generalized linear model

Achamyeleh Birhanu Teshale[1]*, Getayeneh Antehunegn Tesema[1], Misganaw Gebrie Worku[2], Yigizie Yeshaw[1,3], Zemenu Tadesse Tessema[1]

1 Department of Epidemiology and Biostatistics, Institute of Public Health, College of Medicine and Health Sciences, University of Gondar, Gondar, Ethiopia, 2 Department of Human Anatomy, College of Medicine and Health Science, School of Medicine, University of Gondar, Gondar, Ethiopia, 3 Department of Physiology, School of Medicine, College of Medicine and Health Sciences, University of Gondar, Gondar, Ethiopia

* achambir08@gmail.com

## Abstract

### Background

Anemia in women of reproductive age is a major public health challenge for low- and middle-income countries with a long-term negative impact on the health of women, their children, and the economic growth of the society. Even though the world health organization targeted a 50% global reduction of anemia among women of reproductive age by 2025, with the current trend it is unlikely to achieve this goal.

### Objective

This study aimed to assess the prevalence and associated factors of anemia among women of reproductive age in eastern Africa.

### Methods

A secondary data analysis, using demographic and health survey (DHS) data of 10 eastern African countries, was conducted. For our study, a total weighted sample of 101524 women of reproductive age was used. We employed a multilevel mixed-effects generalized linear model (using Poisson regression with robust error variance). Both unadjusted and adjusted prevalence ratios with their 95% confidence interval were reported.

### Results

The prevalence of anemia in eastern Africa was 34.85 (95%CI: 34.56–35.14) ranging from 19.23% in Rwanda to 53.98% in Mozambique. In the multivariable multilevel analysis, being older age, having primary and above education, being from households with second to highest wealth quantiles, being currently working, not perceiving distance as a big problem, use of modern contraceptive methods, and rural residence was associated with a lower

**Data Availability Statement:** All relevant data are within the manuscript.

**Funding:** The author(s) received no specific funding for this work.

**Competing interests:** The authors have declared that no competing interests exist.

**Abbreviations:** aPR, Adjusted Prevalence Ratio; CI, Confidence Interval; DHS, Demographic and Health Surveys; ICC, Intra-class Correlation Coefficient; MOR, Median Odd Ratio; PCV, Proportional Change in Variance; uPR, unadjusted Prevalence Ratio; WHO, World Health organization.

prevalence of anemia. While, being married and divorced/separated/widowed, women from female-headed households, women from households with unimproved toilet facility and unimproved water source, ever had of a terminated pregnancy, having high parity, and being from large household size was associated with a higher prevalence of anemia.

## Conclusion

The prevalence of anemia in eastern Africa was relatively high. Both individual level and community level factors were associated with the prevalence of anemia in women of reproductive age. Therefore, giving special attention to those women who are at a higher prevalence of anemia such as younger women, those who are from households with low socioeconomic status, unimproved toilet facility, and source of drinking water, as well as pregnant women could decrease anemia in women of reproductive age.

## Background

Anemia is a condition in which the number of healthy red blood cells/ hemoglobin (Hgb) level (and consequently their oxygen-carrying capacity) is insufficient to meet the body's physiologic needs [1, 2]. Anemia affects more than 500 million women of reproductive age globally and it is a major public health challenge for low- and middle-income countries (LMICs) with a long-term negative effect on the health of women, their children, and the economic growth [3–5].

Anemia in women of reproductive age has a tremendous effect on the women such as; loss of productivity due to reduced work capacity, cognitive impairment, increased susceptibility to infections due to its effect in immunity, stillbirth/miscarriage, and maternal mortality [6–10]. Besides, anemia in women of reproductive age can result in poor feto-neonatal outcomes such as preterm birth, low birth weight, depletion of the iron stores of the newborn, and in general, it may end up with infant/child mortality [9–13].

The most common type of anemia worldwide is nutritional anemia mainly due to iron, folate, and vitamin B12 deficiencies. Iron deficiency anemia is the most common cause of anemia, with over 50% of anemia are due to iron deficiency [14–16]. Iron deficiency is common in women of reproductive age because of their high demand for iron during pregnancy, lactation, menstrual blood loss, and nutritional deficiencies during their reproductive cycle [9, 17].

Globally, in 2011, the prevalence of anemia in pregnant women was 38% and in non-pregnant women was 29% [18]. Even though anemia affects all countries, it mostly affects LMICs especially Asian and Sub-Saharan African countries which accounts for 89% of the anemia burden [19]. In eastern Africa, the prevalence of anemia in women of reproductive age is higher, which ranges from 19.2% in Rwanda to 49% in Zambia [20–26].

According to different studies done worldwide; age [27, 28], educational level [29–31], occupation [32, 33], marital status [20, 34, 35], wealth status [20, 21, 29, 30, 36], sex of household head [32, 37, 38], media exposure [39–41], body mass index [20, 29, 35, 42], type of toilet facility and source of drinking water [21, 29], ever had of terminated pregnancy [39, 43, 44], parity [36, 45], household size [46, 47], modern contraceptive use [20, 48], current pregnancy status [21, 28, 30, 35, 45], currently breastfeeding [30, 39], residence [49], and community literacy level [32] are associated with anemia in women of reproductive age.

The world health organization (WHO) puts anemia as a public health problem if it is greater than 5% [50], but most of the studies indicated above revealed that the prevalence of anemia in women of reproductive age is above 20%. Also, WHO has set a global target of achieving a 50% reduction of anemia in women of reproductive age by 2025, even though it is unlikely to achieve this plan with the current trend [51]. Therefore, this study aimed to assess the prevalence and associated factors of anemia in women of reproductive age. We hypothesized that the prevalence of anemia in women of reproductive age in eastern Africa is high and different factors are associated with anemia development. The findings of this study will have an advantage in informing policymakers and program planners for making better decisions and plan appropriate intervention strategies to tackle this major public health problem and achieve the plan set by the WHO.

## Methods

### Data source, sampling technique, and population

This study was based on the current 10 Demographic and Health Surveys (DHS) conducted between 2008 and 2018 in Eastern African countries; Burundi, Ethiopia, Malawi, Mozambique, Rwanda, Tanzania, Uganda, Zimbabwe, Madagascar, and Zambia, since the rest two east African countries (Kenya and Comoros) had no recorded anemia or hemoglobin level in the data set, after appending the data sets. The DHS used the stratified cluster sampling technique by using their respective population and housing census as a sampling frame [52]. For this study, we used a weighted sample of 101524 women of reproductive age.

### Variables of the study

**Dependent variable.** This study was based on altitude adjusted hemoglobin level, which was already provided in the DHS data. The outcome variable was anemia level, which was measured based on women's pregnancy status as; if pregnant a hemoglobin value <11 g/dL, and if non-pregnant a hemoglobin value <12 g/dL is considered anemic. In addition, based on severity anemia was classified as severe (if Hgb value <7 g/dL), and moderate (if Hgb value 7–9.9 g/dL) in women of reproductive age and mild (if Hgb level is 10.0–10.9 g/dL) in pregnant women and non-pregnant women (if Hgb level is 10.0–11.9 g/dL). For this study, we re-categorized anemia level as anemic coded as "1" and non-anemic coded as "0" from the previous classifications (no, mild, moderate, and severe) since there were very small numbers of cases in the categories of severe and moderate anemia.

**Independent variables.** After reviewing of literature, both individual and community level explanatory variables were considered. Individual level variables included were; age of the respondent, educational level, occupation, marital status, wealth status, sex of household head, media exposure (constructed from three variables; frequency of listening radio, frequency of watching television, and frequency of reading newspaper), type of toilet facility, source of drinking water, ever had of a terminated pregnancy, parity, household size, perception of distance from the health facility, modern contraceptive use, current pregnancy status, and breastfeeding. Residence, community literacy level, and community poverty level were included as community level variables. Community literacy level and poverty level were created by aggregating individual level variables at cluster/community level since these variables are found to be factors for anemia and not directly found in the DHS.

**Community poverty level**: is the proportion of women in the community who have low household wealth quantiles (lowest and second quantiles).

**Community literacy level**: is the proportion of women in the community who have primary and above educational levels.

To categorize as low and high we used national median value ($<50$ as low and $\geq$ as high) since these variables were not normally distributed.

## Data management and statistical analysis

Extraction, further coding, and both descriptive and analytical analysis were carried out using STATA version 14 software. Weighting was done throughout the analysis to take into account/adjust disproportional sampling and non-response as well as to restore the representativeness of the sample so that the total sample looks like the country's actual population. Descriptive analysis was carried out using frequencies and percentages. The multilevel model was fitted due to the hierarchical nature of the DHS data. In our study, since the prevalence of anemia was high and the outcome was binary, we employed a multilevel mixed-effects generalized linear model (using Poisson regression with robust error variance). Besides, the Intraclass Correlation Coefficient (ICC), the Proportional Change in Variance (PCV), and the Median odds Ratio (MOR) were reported to check whether there was a clustering effect/variability. Bi variable analysis was first done to select variables for multivariable analysis and variables with p-value $<0.20$ in the bivariable analysis were eligible for the multivariable analysis. While doing the multilevel analysis four models; the null model (containing outcome variable only), model I (containing only individual level variables), model II (containing community level variables only), and model III (incorporating both individual and community level variables simultaneously) were fitted. Model comparison was done using deviance and unadjusted and adjusted prevalence ratio (PR) with 95% confidence interval (CI) was reported for the best-fitted model. Finally, variables with p-value $<0.05$ in the multivariable multilevel regression analysis were considered to be significant factors associated with the prevalence of anemia in women of reproductive age.

## Ethical consideration

Since this is a secondary analysis of DHS data, ethical approval was not necessary. But we registered and requested access to these DHS datasets from DHS on-line archive and received approval to access and download the data files.

## Results

### Sociodemographic characteristics

For this study, we used a total weighted sample of 101524 women of reproductive age with the majority (14.70%) of the participants from Ethiopia. The median age of the study participants was 28 (IQR = 20–35) years with the majority (21.97%) between 15 to 19 years. Most (47.86%) of our study participants had primary education and 62.18% of them had got married. Around one fourth (24.23%) of participants were from households with the highest wealth quintile and 56.75% of respondents had a job/ currently working. Regarding sex of household head and media exposure, about 70.45% and 66.79% of respondents were from male-headed households and had media exposure respectively. More than two-thirds (69.85%) of participants were from households with an improved water source and only 44.71% of participants were from households with improved toilet facility. Regarding parity, 34.68% and 26.90% of respondents were multiparous and nulliparous respectively. Most (59.02%) of respondents did not perceive distance from the health facility as a big problem and around three fourth (71.56%) of respondents were rural dwellers (Table 1).

**Table 1. Sociodemographic characteristics of respondents.**

| Variables | Frequency | Percentage |
|---|---|---|
| Country | | |
| Burundi | 8587 | 8.46 |
| Ethiopia | 14923 | 14.70 |
| Madagascar | 8308 | 8.18 |
| Malawi | 7933 | 7.81 |
| Mozambique | 13571 | 13.37 |
| Rwanda | 6680 | 6.58 |
| Tanzania | 13063 | 12.87 |
| Uganda | 5988 | 5.90 |
| Zambia | 13234 | 13.04 |
| Zimbabwe | 9236 | 9.10 |
| Age (years) | | |
| 15–19 | 22301 | 21.97 |
| 20–24 | 18900 | 18.62 |
| 25–29 | 17163 | 16.91 |
| 30–34 | 14632 | 14.41 |
| 35–39 | 12165 | 11.98 |
| 40–44 | 9286 | 9.15 |
| 45–49 | 7077 | 6.97 |
| Educational level | | |
| No education | 21503 | 21.18 |
| Primary | 48585 | 47.86 |
| Secondary | 27817 | 27.40 |
| Higher | 3619 | 3.56 |
| Marital status | | |
| Never married | 26233 | 25.84 |
| Married | 63127 | 62.18 |
| Divorced/widowed/separated | 12164 | 11.98 |
| Occupation | | |
| Working | 57612 | 56.75 |
| Not working | 43912 | 43.25 |
| Household wealth quintile | | |
| Lowest | 18306 | 18.03 |
| Second | 18651 | 18.37 |
| Middle | 18940 | 18.66 |
| Fourth | 21025 | 20.71 |
| Highest | 24602 | 24.23 |
| Sex of household head | | |
| Male | 71520 | 70.45 |
| Female | 30004 | 29.55 |
| Media exposure | | |
| Yes | 67811 | 66.79 |
| No | 33713 | 33.21 |
| Type of toilet facility | | |
| Improved | 45387 | 44.71 |
| Unimproved | 56137 | 55.29 |
| Source of drinking water | | |

(*Continued*)

**Table 1.** (Continued)

| Variables | Frequency | Percentage |
|---|---|---|
| Improved | 70913 | 69.85 |
| Unimproved | 30611 | 30.15 |
| Ever had of a terminated pregnancy | | |
| Yes | 11622 | 11.45 |
| No | 89902 | 88.55 |
| Parity | | |
| None | 27310 | 26.90 |
| Primiparous | 14851 | 14.63 |
| Multiparous | 35205 | 34.68 |
| Grand multiparous | 24158 | 23.80 |
| Household size | | |
| 1–2 | 7899 | 7.78 |
| 3–5 | 44659 | 43.99 |
| 6 and above | 48966 | 48.23 |
| Distance from the health facility | | |
| Big problem | 41604 | 40.98 |
| Not a big problem | 59920 | 59.02 |
| Modern contraceptive use | | |
| Yes | 27905 | 27.49 |
| No | 73619 | 72.51 |
| Currently pregnant | | |
| Yes | 8511 | 8.38 |
| No/unsure | 93013 | 91.62 |
| Currently breastfeeding | | |
| Yes | 27690 | 27.27 |
| No | 73834 | 72.73 |
| Residence | | |
| Urban | 28871 | 28.44 |
| Rural | 72653 | 71.56 |
| Community poverty level | | |
| Low | 50845 | 50.08 |
| High | 50679 | 49.92 |
| Community literacy level | | |
| Low | 52180 | 51.40 |
| High | 49344 | 48.60 |

## Prevalence of anemia among women of reproductive age in eastern Africa

The prevalence of anemia in reproductive age women in eastern Africa was 34.85 (95%CI: 34.56–35.14) with huge variation between countries ranged from 19.23% in Rwanda to 53.98% in Mozambique (Fig 1). Fig 2 shows the spatial distribution of anemia in eastern Africa with the red dots indicating areas with the highest prevalence of anemia.

## Random effects analysis/variability

The community level variability was assessed by both ICC and MOR. As shown in Table 2 the ICC and the MOR values in the null model, which was 6% and 1.54 respectively supports that

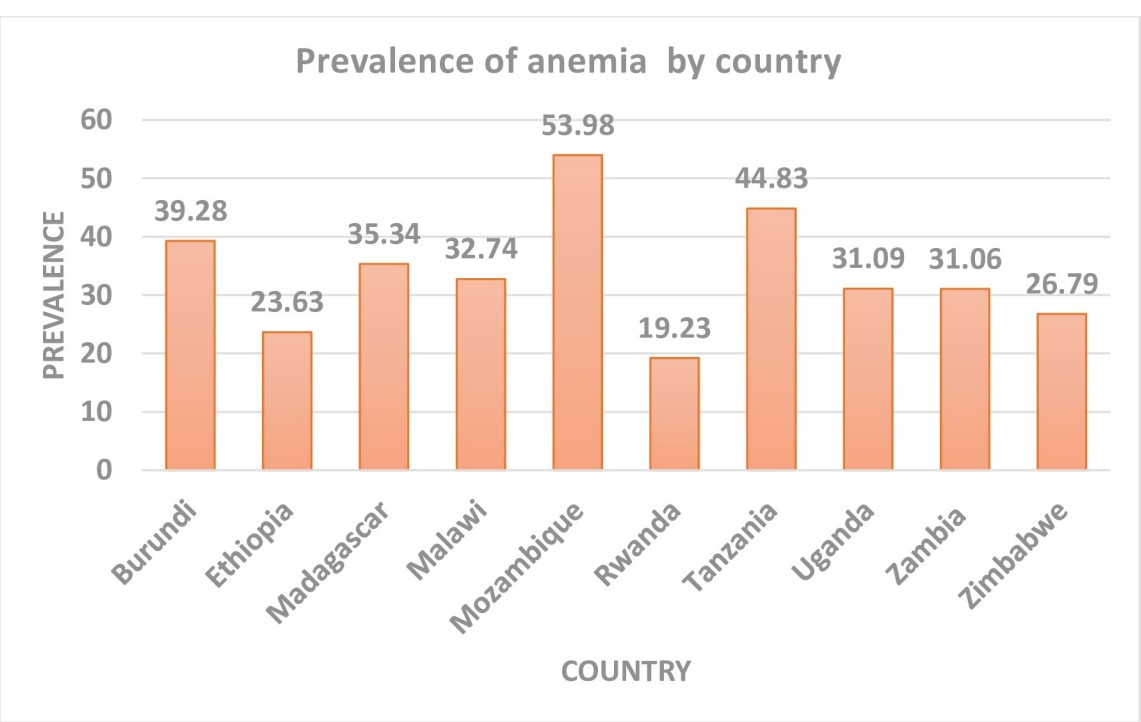

**Fig 1. Prevalence of anemia in eastern Africa showing great variation between countries.**

there was clustering or community level variability of anemia. In addition, the highest PCV in the final model (model 3) revealed that higher proportions of the variation of anemia in women of reproductive age were explained by both individual level and community level factors. Regarding model comparison, deviance was used to select the best fit model among the four models. The model with the lowest deviance, the final model (model III) which incorporates both individual and community level factors simultaneously, was selected as the best-fitted model and we used it to assess the factors associated with anemia among women of reproductive age in eastern Africa (Table 2).

## Factors associated with anemia among women of reproductive age in eastern Africa

All variables (both individual level and community level variables) had p-value <0.20 in the bivariable analysis and were eligible for multivariable analysis. In the multivariable analysis; the individual level factors such as age, education, marital status, occupation, household wealth status, sex of household head, type of toilet facility, source of drinking water, ever had of a terminated pregnancy, parity, household size, perception of distance from the health facility, and pregnancy status were significant determinants of anemia among women of reproductive age. Among community level factors, residence was significantly associated with anemia in women of reproductive age (Table 3).

Being in the older age group was associated with a lower prevalence of anemia as compared to the age group 15–19 years. The prevalence of anemia was 8%, 13%, and 21% lower in women had primary, secondary, and higher education, respectively, as compared with women with no formal education. The prevalence of anemia was 9% and 15% higher in women who were married and divorced/separated/widowed, respectively, as compared with those who

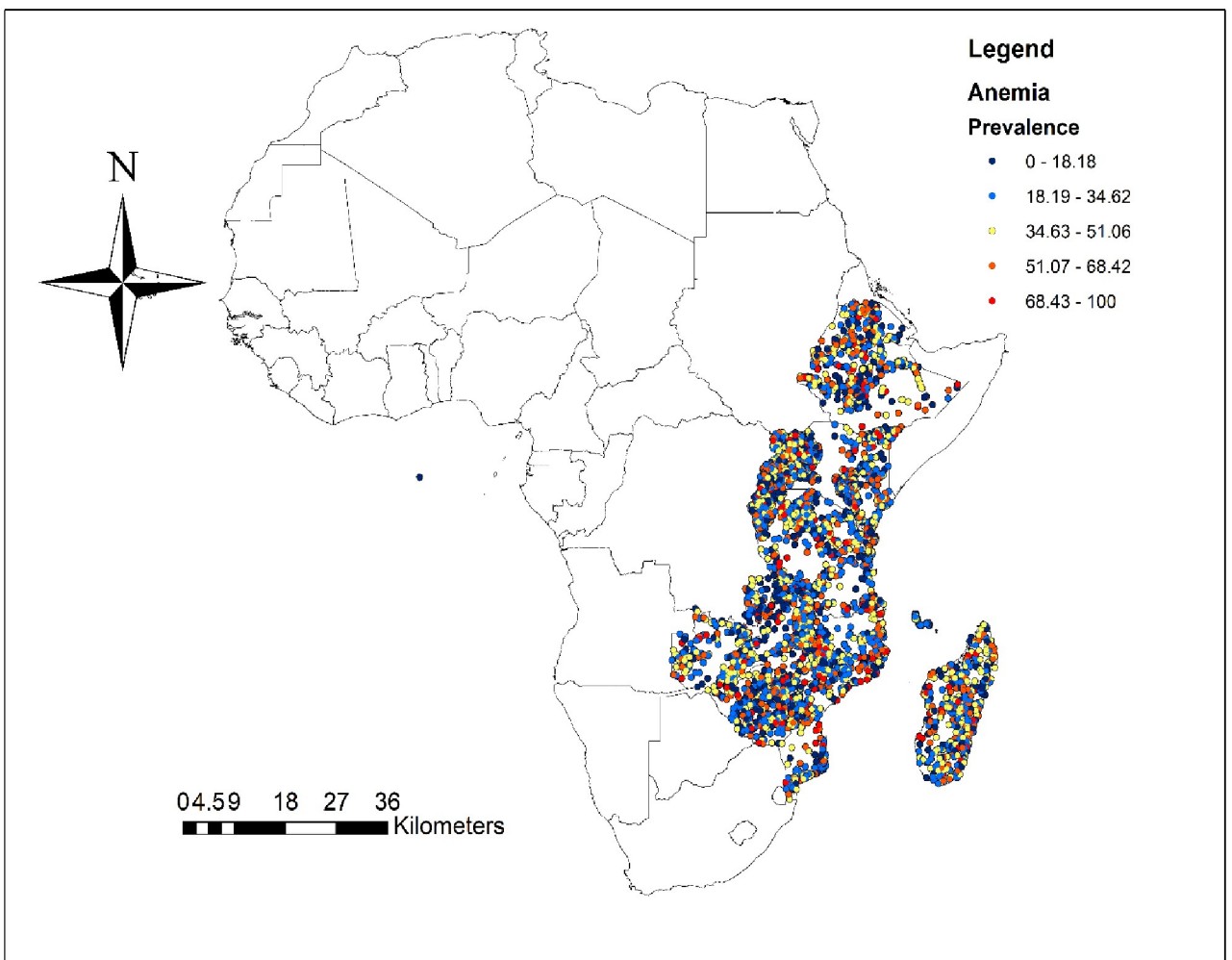

**Fig 2. Spatial distribution of anemia among women of reproductive age in eastern Africa.**

were never married. The prevalence of anemia in currently working women was 3% lower as compared with their counterparts. Regarding household wealth quantiles, being women from second, middle, fourth, and highest household wealth quantiles was associated with 6%, 7%, 7%, and 11% lower prevalence of anemia as compared to those who were from the lowest household wealth quintile. Being women from female-headed households was associated with 5% higher prevalence of anemia as compared with male-headed households. Women from

**Table 2. Community level variability and model fitness for assessment of anemia among women of reproductive age in eastern Africa.**

| Parameter | Null model | Model I | Model II | Model III |
|---|---|---|---|---|
| Community level variance | 0.21 | 0.20 | 0.16 | 0.15 |
| ICC | 0.06 | 0.05 | 0.06 | 0.04 |
| MOR | 1.54(1.48–1.58) | 1.53(1.47–1.58) | 1.46(1.41–1.52) | 1.44(1.39–1.51) |
| PCV (%) | Reference | 5% | 24% | 29% |
| **Model fitness** | | | | |
| Deviance (-2LL) | 145704 | 144310 | 145622 | 144228 |

**Table 3. Bi variable and multivariable multilevel regression analysis to assess factors associated with anemia among women of reproductive age in eastern Africa.**

| Variables | Anemia | | Prevalence ratio(PR) | |
|---|---|---|---|---|
| | No | yes | uPR (95%CI) | aPR (95%CI) |
| Age (years) | | | | |
| 15–19 | 14467 | 7834 | 1.00 | 1.00 |
| 20–24 | 12365 | 6535 | 0.98(0.94–1.03) | 0.96(0.93–0.99) * |
| 25–29 | 11472 | 5610 | 0.96(0.93–0.99) | 0.92(0.89–0.96) * |
| 30–34 | 9692 | 4910 | 0.96(0.94–0.99) | 0.91(0.88–0.95) * |
| 35–39 | 7779 | 4386 | 1.04(1.01–1.07) | 0.97(0.93–0.99) * |
| 40–44 | 5890 | 3396 | 1.04(1.01–1.08) | 0.96(0.91–0.99) * |
| 45–49 | 4477 | 2600 | 1.03(0.99–1.07) | 0.92(0.87–0.97) * |
| Educational level | | | | |
| No education | 12921 | 8582 | 1.00 | 1.00 |
| Primary | 31371 | 17214 | 0.87(0.4–0.89) | 0.92(0.90–0.95) * |
| Secondary | 19190 | 8627 | 0.77(0.75–0.79) | 0.87(0.84–0.90) * |
| Higher | 2660 | 959 | 0.66(0.62–0.72) | 0.79(0.74–0.83) * |
| Marital status | | | | |
| Never married | 17673 | 8560 | 1.00 | 1.00 |
| Married | 41081 | 22046 | 1.07(1.05–1.10) | 1.09(1.05–1.12) * |
| Divorced/widowed/separated | 7388 | 4776 | 1.18(1.15–1.22) | 1.15(1.11–1.19) * |
| Occupation | | | | |
| Working | 28616 | 15296 | 0.96(0.94–0.98) | 0.97(0.95–0.99) * |
| Not working | 37526 | 20086 | 1.00 | 1.00 |
| Household wealth quintile | | | | |
| lowest | 10864 | 7442 | 1.00 | 1.00 |
| second | 11928 | 6723 | 0.90(0.87–0.93) | 0.94(0.91–0.97) * |
| Middle | 12376 | 6564 | 0.87(0.84–0.90) | 0.93(0.90–0.96) * |
| Fourth | 14090 | 6935 | 0.85(0.82–0.88) | 0.93(0.90–0.97) * |
| Highest | 16884 | 7718 | 0.78(0.75–0.81) | 0.89(0.85–0.93) * |
| Sex of household head | | | | |
| Male | 47008 | 24512 | 1.00 | 1.00 |
| Female | 19134 | 10870 | 1.04(1.02–1.06) | 1.05(1.02–1.07) * |
| Media exposure | | | | |
| Yes | 44647 | 23164 | 0.91(0.89–0.94) | 1.02(1.01–1.04) |
| No | 21495 | 12218 | 1.00 | 1.00 |
| Type of toilet facility | | | | |
| Improved | 30689 | 14698 | 1.00 | 1.00 |
| Unimproved | 35453 | 20684 | 1.16(1.13–1.19) | 1.05(1.02–1.07) * |
| Source of drinking water | | | | |
| Improved | 47338 | 23573 | 1.00 | 1.00 |
| Unimproved | 18804 | 11807 | 1.15(1.12–1.18) | 1.04(1.01–1.07) * |
| Ever had of a terminated pregnancy | | | | |
| Yes | 7365 | 4257 | 1.07(1.04–1.09) | 1.06(1.03–1.09) * |
| No | 58777 | 31125 | 1.00 | 1.00 |
| Parity | | | | |
| None | 18252 | 9057 | 1.00 | 1.00 |
| Primiparous | 9435 | 5416 | 1.07(1.05–1.11) | 1.11(1.07–1.15) * |
| Multiparous | 23216 | 11989 | 1.03(1.01–1.05) | 1.07(1.03–1.12) * |
| Grand multiparous | 15239 | 8919 | 1.11(1.09–1.14) | 1.06(1.01–1.11) * |

(*Continued*)

**Table 3.** (*Continued*)

| Variables | Anemia | | Prevalence ratio(PR) | |
|---|---|---|---|---|
| | No | yes | uPR (95%CI) | aPR (95%CI) |
| Household size | | | | |
| 1–2 | 5075 | 2824 | 1.00 | 1.00 |
| 3–5 | 29609 | 15050 | 0.95(0.91–0.98) | 0.98(0.94–1.01) |
| 6 and above | 31458 | 17588 | 1.01(0.98–1.05) | 1.05(1.01–1.09) * |
| Distance from the health facility | | | | |
| Big problem | 26239 | 15365 | 1.00 | 1.00 |
| Not a big problem | 39903 | 20017 | 0.90(0.88–0.92) | 0.96(0.94–0.98) * |
| Modern contraceptive use | | | | |
| Yes | 20584 | 7321 | 0.70(0.68–0.71) | 0.71(0.69–0.73) * |
| No | 45558 | 28061 | 1.00 | 1.00 |
| Currently pregnant | | | | |
| Yes | 4922 | 3590 | 1.24(1.20–1.27) | 1.11(1.08–1.13) * |
| No/unsure | 61221 | 31792 | 1.00 | 1.00 |
| Currently breastfeeding | | | | |
| Yes | 17613 | 10077 | 1.06(1.04–1.08) | 1.02(0.99–1.04) |
| No | 48529 | 25305 | 1.00 | 1.00 |
| Residence | | | | |
| Urban | 19069 | 9802 | 1.00 | 1.00 |
| Rural | 47073 | 25580 | 1.10(1.07–1.14) | 0.94(0.90–0.98) * |
| Community poverty level | | | | |
| Low | 33499 | 17346 | 1.00 | 1.00 |
| High | 32643 | 18036 | 1.03(0.99–1.05) | 0.97(0.95–1.01) |
| Community literacy level | | | | |
| Low | 33626 | 18554 | 1.00 | 1.00 |
| High | 32516 | 16828 | 0.94(0.92–0.97) | 0.98(0.95–1.01) |

aPR = adjusted Prevalence Ratio, uPR = unadjusted Prevalence Ratio,

* = p value<0.05.

households with unimproved toilet facility and unimproved sources of water had 5% and 4% higher prevalence of anemia, respectively than their counterparts. Women with ever had of a terminated pregnancy had 6% higher prevalence of anemia as compared with their counterparts. Regarding parity of the respondent, primiparous, multiparous, and grand multiparous women had 11%, 7%, and 6% higher prevalence of anemia respectively, as compared to nulliparous women. Being women from larger household size (six and above) was associated with 5% higher prevalence of anemia as compared to those from households with a household size of one to two. Being a woman not perceiving distance from the health facility as a big problem was associated with 4% lower prevalence of anemia as compared to their counterparts. Using modern contraceptive methods was associated with 29% lower prevalence of anemia than women who were not using modern contraceptive methods. Being currently pregnant was associated with 11% higher prevalence of anemia as compared to non-pregnant women. Moreover, being from a rural area was associated with 6% lower prevalence of anemia as compared to urban areas (Table 3).

## Discussion

Anemia is a major public health problem in reproductive age women because of their high demand for iron during pregnancy, lactation, menstrual bleeding, and nutritional deficiency during their reproductive cycle [9]. This study assessed the prevalence of anemia and its associated factors among women of reproductive age in eastern African countries. In this study, the prevalence of anemia among women of reproductive age was 34.85 (95%CI: 34.56–35.14) and this is consistent with studies done in India and Nepal [53, 54]. Anemia prevalence in this study was higher than studies done in Brazil [27], Iran [36], Thailand [55], Turk [56], and Timor-Lest [31]. But this prevalence of anemia is lower than studies done in Nepal [34], Myanmar [35], Democratic Republic of Congo [28], India [57], and Vietnam [58]. This difference in anemia prevalence between countries may be due to the variation in geographical, cultural, and dietary-related factors between countries. In addition, the high prevalence of anemia among women in the countries of eastern Africa may be attributable to their social and biological susceptibility to anemia. Moreover, in developing countries especially in Eastern African countries, access to iron-rich food is inadequate due to their poor socioeconomic status, inadequate health care accesses, and utilization and this may result in anemia. In addition, this regional variation of anemia might be associated with the variation in the distribution and prevalence of communicable disease that commonly affect developing countries like eastern African countries.

Consistent with studies conducted elsewhere [21, 28–39, 45–49, 59], in our study, being older age, having primary and above education, being from households with second to highest wealth quintiles, being currently working, not perceiving distance as not a big problem, use of modern contraceptive methods, and rural residence was associated with a lower prevalence of anemia. While, being married and divorced/separated/widowed, women from female-headed households, women from households with unimproved toilet facility and unimproved water source, ever had of a terminated pregnancy, having high parity, and from large household size was associated with a higher prevalence of anemia.

In the study, the prevalence of anemia was lower among women who had primary and above education compared with those women who had no formal education. This finding is congruent with studies done in Ethiopia [30], Rwanda [29], Timor-Leste [31]. This might be because educated mothers usually eat a variety of foods such as vitamins and minerals which might lead to a reduction in nutritional deficiency anemia. In addition, obtaining education may help women adopt appropriate lifestyle patterns such as better health-seeking habits as well as hygiene practices that can prevent women from getting anemia. Consistent with other studies conducted in different settings [20, 21, 29, 30, 36, 60], in this study, being from second to highest household wealth quantiles were associated with lower prevalence of anemia as compared with women from households with lowest quantile. This could be due to improved socioeconomic status is associated with healthy nutrition, lower infection/morbidity, and increased access and utilization of medical health services [59, 61, 62]. In addition, it might be because of women from high socioeconomic status could purchase variety (both in quality and quantity) of foods.

The study at hand also revealed that being from households with unimproved toilet facility and unimproved sources of drinking water associated with a higher prevalence of anemia and this is in line with studies conducted Uganda and Ruanda [21, 29, 60]. This might be because women with unimproved toilet facility and unimproved sources of drinking water are at risk of both waterborne and foodborne diseases which might in turn, increases the risk of anemia. Moreover, these groups of women are at risk of getting helminthic infections such as hookworm, which is the most common cause of anemia in poor sanitary conditions. We also found

that being pregnant was associated with a higher prevalence of anemia as compared with those who were not pregnant. This is in concordance with studies done in Jourdan [45], Democratic Republic of Congo [28], Rwanda [35], Ethiopia [30], and Uganda [21, 60]. This is due to the fact that pregnant women have an increased demand for iron to sustain her baby's development. Another possible explanation will be during pregnancy nutritional deficiencies, bacterial and parasitic infections, and genetic disorders of the red blood cells such as thalassemia is common, which could eventually lead to anemia [63]. Our study also revealed that modern contraceptive use was associated with anemia in women of reproductive age. Using modern contraceptive methods reduces the prevalence of anemia and this is in concordance with different studies [20, 32, 48]. This is because women who used modern contraceptive methods prevent complications related to pregnancy and childbirth, which could eventually reduce the prevalence of anemia due to recurrent blood loss. Another plausible explanation will be using modern contraception methods (especially hormonal contraceptive methods) could minimize the menstrual bleeding and reduce their susceptibility to anemia [64, 65].

Our study also found distance from the health facility as a significant anemia-related factor in which women who consider distance from the health facility as a major problem were at higher risk of anemia. This might be due to the fact that women who were far from the nearest facility cannot access maternal health services timely such as iron and folate supplementation during pregnancy, modern contraceptives as well as other services related to the continuum of care, which all make them susceptive to anemia. Moreover, in this study women from rural areas had lower odds of anemia as compared with those who were from urban areas. This is consistent with studies done in Malawi [49]. It might be because women living in rural areas usually have an increased access and utilization of Teff (a type of crop used to make Enjera, a traditional food in Ethiopia) and other iron-containing foods that can lead to a reduction in the risk of nutritional anemia.

This study was based on a multicounty analysis with a large sample size and appropriate statistical analysis considering the hierarchical nature of the DHS data. Therefore, we authors strongly believe that it provides more precise and generalizable findings that can be used by policymakers and program planners to design intervention strategies for the problem at both individual and community levels. However, this study was not without limitations. Due to the cross-sectional nature of the DHS data, we are unable to establish a cause and effect relationship between independent variables and anemia. Moreover, since the study was based on information available on the surveys, other confounders such as infections (such as malaria, intestinal parasites, and HIV/AIDS) were not adjusted.

## Conclusion

The prevalence of anemia in eastern Africa was relatively high. Both individual level and community level factors were associated with the development of anemia in women of reproductive age. Giving special attention for those groups of women who had a higher prevalence of anemia such as younger women, uneducated women, those who are from households with low socioeconomic status, unimproved toilet facility and source of drinking water is recommended.

## Acknowledgments

We would like to acknowledge the MEASURE DHS program which helps us to access and use the data set for 2016 EDHS.

## Author Contributions

**Conceptualization:** Achamyeleh Birhanu Teshale, Getayeneh Antehunegn Tesema, Misganaw Gebrie Worku, Yigizie Yeshaw, Zemenu Tadesse Tessema.

**Data curation:** Achamyeleh Birhanu Teshale, Getayeneh Antehunegn Tesema, Misganaw Gebrie Worku, Yigizie Yeshaw, Zemenu Tadesse Tessema.

**Formal analysis:** Achamyeleh Birhanu Teshale, Getayeneh Antehunegn Tesema, Misganaw Gebrie Worku, Yigizie Yeshaw, Zemenu Tadesse Tessema.

**Investigation:** Achamyeleh Birhanu Teshale, Getayeneh Antehunegn Tesema, Misganaw Gebrie Worku, Yigizie Yeshaw, Zemenu Tadesse Tessema.

**Methodology:** Achamyeleh Birhanu Teshale, Getayeneh Antehunegn Tesema, Misganaw Gebrie Worku, Yigizie Yeshaw, Zemenu Tadesse Tessema.

**Resources:** Achamyeleh Birhanu Teshale, Getayeneh Antehunegn Tesema, Misganaw Gebrie Worku, Yigizie Yeshaw, Zemenu Tadesse Tessema.

**Software:** Achamyeleh Birhanu Teshale, Getayeneh Antehunegn Tesema, Misganaw Gebrie Worku, Yigizie Yeshaw, Zemenu Tadesse Tessema.

**Validation:** Achamyeleh Birhanu Teshale, Getayeneh Antehunegn Tesema, Misganaw Gebrie Worku, Yigizie Yeshaw, Zemenu Tadesse Tessema.

**Visualization:** Achamyeleh Birhanu Teshale, Getayeneh Antehunegn Tesema, Misganaw Gebrie Worku, Yigizie Yeshaw, Zemenu Tadesse Tessema.

**Writing – original draft:** Achamyeleh Birhanu Teshale, Getayeneh Antehunegn Tesema, Misganaw Gebrie Worku, Yigizie Yeshaw, Zemenu Tadesse Tessema.

**Writing – review & editing:** Achamyeleh Birhanu Teshale, Getayeneh Antehunegn Tesema, Misganaw Gebrie Worku, Yigizie Yeshaw, Zemenu Tadesse Tessema.

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
