## [Decision Letter · Decision Letter 0]

14 Jul 2020

PONE-D-20-18978

Prevalence and associated factors of anemia among reproductive-aged women in Eastern Africa; a multilevel analysis of the recent DHS data.

PLOS ONE

Dear Dr. Teshale,

Thank you for submitting your manuscript to PLOS ONE. After careful consideration, we feel that it has merit but does not fully meet PLOS ONE’s publication criteria as it currently stands. Therefore, we invite you to submit a revised version of the manuscript that addresses the points raised during the review process.

SPECIFIC ACADEMIC EDITOR COMMENTS: An expert in the field handled your manuscript. We really appreciate their time and efforts. Although interest was found in your study, some major concerns and comments reduced this enthusiasm. There are comments about revising several statements to increase clarity; additional details about the methods need to be provided; there are questions about the data analyses; and the discussion should be rewritten to better focus on the main and novel findings. Please address ALL of the reviewer's comments in your revised manuscript.

We look forward to receiving your revised manuscript.

Kind regards,

Frank T. Spradley

Academic Editor

PLOS ONE

Journal Requirements:

Reviewers' comments:

Reviewer's Responses to Questions

**Comments to the Author**

1. Is the manuscript technically sound, and do the data support the conclusions?

Reviewer #1: Yes

2. Has the statistical analysis been performed appropriately and rigorously? 

Reviewer #1: Yes

3. Have the authors made all data underlying the findings in their manuscript fully available?

Reviewer #1: Yes

4. Is the manuscript presented in an intelligible fashion and written in standard English?

Reviewer #1: Yes

5. Review Comments to the Author

Reviewer #1: The manuscript by Teshale and colleagues explored the prevalence and associated factors of anemia among reproductive-aged women in Eastern Africa. The authors used a weighted and large sample of DHS data. The authors found a high prevalence of anemia in east Africa with spatial variations. The authors delineated possible associated factors using appropriate statistical techniques. The following suggestions can strengthen the manuscript:

Abstract

1. Change “Anemia in reproductive age women” to “Anemia in women of reproductive age."

2. For culture appropriateness, “restrain to “low- and middle-income countries" as opposed to "developing countries."

3. Start a new sentence with “This study aimed to”

4. Change” …using the nine eastern African countries DHS data” to using DHS data of 9 eastern African countries,”

5. Defined DHS at the first mention or avoid abbreviation in the abstract.

6. Delete “and variables with p-value <0.05 in the multivariable analysis were declared as significant determinants of anemia."

Background

1. Line 50: delete "developing."

2. Line 50-57: For coherence, separate the effect of anemia women and their offspring separately.

3. Line 63-64: Add in 2011. As you know, the prevalence should be stated with a time period.

4. Remove "developing countries" to replace them with LMIC.

5. Please clearly state the objective of threw study in the last paragraph of the background before starting line 80.

6. In addition, please state your hypothesis

Method

1. Change "method" to "methods."

2. Cite line 90

3. Please write all “reproductive-age women" to "women of reproductive age."

4. Give the rationales for choosing the confounding factors to adjust for.

5. Line 115: The DHS uses a different categorization of wealth index, please use them as opposed to poorest/poorer

6. Line 124: Since the prevalence of the outcome of interest is very high (35%), the odds ratio can overestimate the prevalence ratio using logistic regression. Although the use of odds ratios is correct, authors should consider prevalence ratios analysis by employing either log-binomial or Poisson regression “see Barros, A. J., & Hirakata, V. N. (2003). Alternatives for logistic regression in cross-sectional studies: an empirical comparison of models that directly estimate the prevalence ratio. BMC medical research methodology, 3(1), 21. See use of log-binomial analysis in anemia prevalence" https://link.springer.com/article/10.1186/s12884-020-03064-x]

7.

8.

9. Describe the methods of sample weighting. Was additional weighting with the population of the country done?

10. Line 134: Authors should explain why started methods of model comparison such as AIC/BIC were not used

Results:

1. I recommend adding a geographical map depicting the distribution of anemia in East Africa.

Discussion

1. As stated on line 247-248, infectious diseases such as malaria and HIV substantially contribute to anemia in the population in LMICs. However, the authors never explored their effect in this study, and yet malaria and HIV data are available in the DHS records. For the recent similar study for the west and central Africa, see Ssentongo, P., Ba, D. M., Ssentongo, A. E., Ericson, J. E., Wang, M., Liao, D., & Chinchilli, V. M. (2020). Associations of malaria, HIV, and coinfection, with anemia in pregnancy in sub-Saharan Africa: a population-based cross-sectional study. BMC Pregnancy and Childbirth, 20(1), 1-11.

2. The should consider cut down the discussion to focus on the main findings and continue to provide 2-3 paragraphs explaining their results in light of other studies. The final section should be the clinical and public health implications and finally, the limitations. The authors do not have to discuss every result.

6. PLOS authors have the option to publish the peer review history of their article (what does this mean?). If published, this will include your full peer review and any attached files.

Reviewer #1: **Yes: **Paddy Ssentongo, MD, MPH

---

## [Author Response · Author response to Decision Letter 0]

10 Aug 2020

Date: August 10, 2020

Author’s response to editor and reviewer

Title: Prevalence and associated factors of anemia among reproductive-aged women in Eastern Africa; a multilevel analysis of the recent DHS data.

Manuscript number: PONE-D-20-18978

Dear editor/reviewer: We have now thoroughly updated the manuscript. We hope the improvements are sufficient and if that is not the case, any suggestions/comments are welcome. Below we present a point-by - point response to the questions raised by the editor & reviewer, to whom we thank for their valuable contributions to this research.

Editor’s comment

Author’s response: Dear editor thank you. We amended our manuscript according to the journal style. 

Reviewer #1: Dr. Paddy Ssentongo 

1. Change “Anemia in reproductive age women” to “Anemia in women of reproductive age."

Author’s response: thank you for the comment we amended to read as “anemia in reproductive age women” throughout the revised document (see the track change) 

2. For culture appropriateness, “restrain to “low- and middle-income countries" as opposed to "developing countries."

Author’s response: Corrected in the revised paper.

3. Start a new sentence with “This study aimed to”

Author’s response: Amended in the revised paper

4. Change” …using the nine eastern African countries DHS data” to using DHS data of 9 eastern African countries,”

Author’s response: Corrected in the revised manuscript 

5. Defined DHS at the first mention or avoid abbreviation in the abstract.

Author’s response: We first mentioned it as “Demographic and Health Survey (DHS)” in the revised manuscript

6. Delete “and variables with p-value <0.05 in the multivariable analysis were declared as significant determinants of anemia."

Author’s response: Thank you. We deleted in the revised manuscript.

7. Line 50: delete "developing."

Author’s response: Deleted 

8. Line 50-57: For coherence, separate the effect of anemia women and their offspring separately (paragraph 2 line 53-58).

Author’s response: Thank you for the important concern you raised. We consider it and put the effects of anemia for women and the newborn/offspring separately. 

9. Line 63-64: Add in 2011. As you know, the prevalence should be stated with a time period.

Author’s response: The time (2011) is added in the revised manuscript.

10. Remove "developing countries" to replace them with LMIC.

Author’s response: Thank you we removed it 

11. Please clearly state the objective of threw study in the last paragraph of the background before starting line 80.

Author’s response: thank you. We consider it in the revised manuscript (see the last paragraph of the background section).

12. In addition, please state your hypothesis

Author’s response: We stated the hypothesis in the revised manuscript (found in the last paragraph of the revised manuscript).

13. Change "method" to "methods."

Author’s response: Changed 

14. Cite line 90

Author’s response: Thank you. We add citation/reference 

15. Please write all “reproductive-age women" to "women of reproductive age."

Author’s response: We amended it throughout the revised manuscript

16. Give the rationales for choosing the confounding factors to adjust for.

Author’s response: Thank you for raising the important concern. We choose the confounding factors based on reviewing of literatures, based on clinical judgment/importance, as well as based on their availability in the DHSs. 

17. Line 115: The DHS uses a different categorization of wealth index, please use them as opposed to poorest/poorer

Author’s response: Thank you for your comment. We amended the terms to read as household wealth quantiles [first, second, middle, fourth, and highest] in the revised manuscript.

18. Line 124: Since the prevalence of the outcome of interest is very high (35%), the odds ratio can overestimate the prevalence ratio using logistic regression. Although the use of odds ratios is correct, authors should consider prevalence ratios analysis by employing either log-binomial or Poisson regression “see Barros, A. J., & Hirakata, V. N. (2003). Alternatives for logistic regression in cross-sectional studies: an empirical comparison of models that directly estimate the prevalence ratio. BMC medical research methodology, 3(1), 21. See use of log-binomial analysis in anemia prevalence" https://link.springer.com/article/10.1186/s12884-020-03064-x]

Author’s response: We really thank you for this important concern and direction. We were trying to fit a multilevel log-binomial model in order to calculate the prevalence ratio (rather than the odds ratio) for a clustered binary outcome using stata [using the command meglm depvar indvar || v001: , family(binomial) link(log) eform ], unfortunately we have got an unexpected error which says link log is not allowed with family Bernoulli. Therefore, we consider a reasonable analytic option that is we use Poisson regression with robust standard errors to overcome the over estimation of the prevalence ratio (an important measure of association if the outcome of interest is common) as stated by [Coutinho L, Scazufca M, Menezes PR, 2008, Zou G, 2004, and Barros, A.J., Hirakata, V.N, 2003].

19. Describe the methods of sample weighting. Was additional weighting with the population of the country done?

Author’s response: Just we appended the data and we weight it using the weighting factor.

20. Line 134: Authors should explain why started methods of model comparison such as AIC/BIC were not used

Author’s response: Since the models were nested we used deviance (the preferable method for model comparison for nested models) 

21. I recommend adding a geographical map depicting the distribution of anemia in East Africa.

Author’s response: Thank you for your comment. We added the map displaying the prevalence of anemia in the revised manuscript (see figure 2).

22. As stated on line 247-248, infectious diseases such as malaria and HIV substantially contribute to anemia in the population in LMICs. However, the authors never explored their effect in this study, and yet malaria and HIV data are available in the DHS records. For the recent similar study for the west and central Africa, see Ssentongo, P., Ba, D. M., Ssentongo, A. E., Ericson, J. E., Wang, M., Liao, D., & Chinchilli, V. M. (2020). Associations of malaria, HIV, and coinfection, with anemia in pregnancy in sub-Saharan Africa: a population-based cross-sectional study. BMC Pregnancy and Childbirth, 20(1), 1-11.

Author’s response: Thank you for your important question. Even though infectious diseases (such as malaria, HIV/AIDS, and intestinal parasites) are known to contribute to anemia, we used the IR (individual women) data set, for this study, in which there was no such variables in most of the surveys. Moreover, they are found as separate file/data and difficult to access and append these data. For example, in Ethiopia information about malaria is not found in the EDHS 2016 (IR data set), rather it is found as separate data (as malaria indicator survey). Therefore, we acknowledge it as limitation of our study (see the limitation section) 

23. The should consider cut down the discussion to focus on the main findings and continue to provide 2-3 paragraphs explaining their results in light of other studies. The final section should be the clinical and public health implications and finally, the limitations. The authors do not have to discuss every result.

Author’s response: Dear reviewer thank you for your important concern. Even though. it is difficult to cut down the discussion section in to 2/3 paragraphs (since we had many findings) we tried to concentrate and discuss the most important variables. We also putted the strength/implications, as well as the limitations of our study in the last paragraph of the discussion section.

Dear reviewer 

We made a re-analysis based on your concern / comment, as well as due to the inclusion of Zambia DHS (2018), which was previously missed in view of the previous Zambian DHS (ZDHS 2013). So, our study was based on 10 East African DHSs with a weighted sample of 101524 women of reproductive age.

References 

Coutinho L, Scazufca M, Menezes PR. Methods for estimating prevalence ratios in cross-sectional studies. Revista de saude publica. 2008 Dec;42(6):992-8.

Zou G. A modified poisson regression approach to prospective studies with binary data. American journal of epidemiology. 2004 Apr 1;159(7):702-6.

Barros, A.J., Hirakata, V.N. Alternatives for logistic regression in cross-sectional studies: an empirical comparison of models that directly estimate the prevalence ratio. BMC Med Res Methodol 3, 21 (2003). https://doi.org/10.1186/1471-2288-3-21

Blumenberg, Cauane. (2017). Re: How can we calculate prevalence ratio from Poisson regression?. Retrieved from: https://www.researchgate.net/post/How_can_we_calculate_prevalence_ratio_from_Poisson_regression/58744263615e27dd6f318ca7/citation/download.

---

## [Decision Letter · Decision Letter 1]

27 Aug 2020

Anemia and its associated factors among women of reproductive age in Eastern Africa: a Multilevel Mixed-effects Generalized Linear Model

PONE-D-20-18978R1

Dear Dr. Teshale,

We’re pleased to inform you that your manuscript has been judged scientifically suitable for publication and will be formally accepted for publication once it meets all outstanding technical requirements.

Kind regards,

Frank T. Spradley

Academic Editor

PLOS ONE

Reviewers' comments:

Reviewer's Responses to Questions

**Comments to the Author**

1. If the authors have adequately addressed your comments raised in a previous round of review and you feel that this manuscript is now acceptable for publication, you may indicate that here to bypass the “Comments to the Author” section, enter your conflict of interest statement in the “Confidential to Editor” section, and submit your "Accept" recommendation.

Reviewer #1: All comments have been addressed

2. Is the manuscript technically sound, and do the data support the conclusions?

Reviewer #1: Yes

3. Has the statistical analysis been performed appropriately and rigorously? 

Reviewer #1: Yes

4. Have the authors made all data underlying the findings in their manuscript fully available?

Reviewer #1: Yes

5. Is the manuscript presented in an intelligible fashion and written in standard English?

Reviewer #1: Yes

6. Review Comments to the Author

Reviewer #1: The authors have addressed all my comments. The manuscript is clearly written and the statistics are robust.

7. PLOS authors have the option to publish the peer review history of their article (what does this mean?). If published, this will include your full peer review and any attached files.

Reviewer #1: **Yes: **Paddy Ssentongo, MD, MPH

---

## [Editor Report · Acceptance letter]

1 Sep 2020

PONE-D-20-18978R1 

Anemia and its associated factors among women of reproductive age in Eastern Africa: a Multilevel Mixed-effects Generalized Linear Model 

Dear Dr. Teshale:

I'm pleased to inform you that your manuscript has been deemed suitable for publication in PLOS ONE. Congratulations! Your manuscript is now with our production department. 

Kind regards, 

on behalf of

Dr. Frank T. Spradley 

Academic Editor

PLOS ONE